# Estimations of Water Volume and External Loading Based on DYRESM Hydrodynamic Model at Lake Dianchi

Rufeng Zhang [1], Liancong Luo [1,*], Min Pan [2], Feng He [2], Chunliang Luo [2], Di Meng [2], Huiyun Li [3,*], Jialong Li [1], Falu Gong [1], Guizhu Wu [1], Lan Chen [1], Jian Zhang [1] and Ting Sun [1]

[1]  Institute for Ecological Research and Pollution Control of Plateau Lakes,
    School of Ecology and Environmental Science, Yunnan University, Kunming 650504, China
[2]  Kunming Dianchi & Plateau Lakes Institute, Kunming 650500, China
[3]  Nanjing Institute of Geography and Limnology, Chinese Academy of Sciences, Nanjing 210008, China
*   Correspondence: billluo@ynu.edu.cn (L.L.); hyli@niglas.ac.cn (H.L.); Tel.: +86-(0)-181-1290-7530 (L.L.);
    +86-(0)-159-0515-7570 (H.L.)

**Abstract:** There are many rivers flowing from complex paths into Lake Dianchi. At present, there is a lack of inflow and water quality monitoring data for some rivers, resulting in limited accuracy of statistical results regarding water volume and external loading estimations. In this study, we used DYRESM to estimate the water volume entering Waihai of Lake Dianchi from 2007 to 2019 without historical hydrological observation data. Then, we combined this information with the monthly monitoring data of water quality to calculate the annual external loading. Our results showed that: (1) DYRESM could effectively capture the extreme changes of water level at Waihai, showing its reliable applicability to Lake Dianchi. (2) The average annual inflow of rivers entering Waihai was about $6.69 \times 10^8$ m$^3$. The fitting relationship between river inflow and precipitation was significant on annual scale ($r = 0.74$), with a higher inner-annual fitting coefficient between them ($r = 0.98$), thus suggesting that precipitation and its caused river inflows are the main water source for Waihai. (3) From 2007 to 2010, the river loadings remained at a high level. They decreased to 2445.44 t (total nitrogen, TN) and 106.53 t (total phosphorus, TP) due to a followed drought in 2011. (4) The river loading had annual variation characteristics. The contribution rates of TN and TP loading in the rainy season were 63% and 67% respectively. (5) Panlong River, Daqing River, Jinjia River, Xinbaoxiang River, Cailian River and Hai River were the main inflow rivers. Their loadings accounted for 81.3% (TN) and 80.3% (TP) of the total inputs. (6) River loadings have gradually reduced and the water quality of Waihai has continually improved. However, Pearson analysis results showed that the water quality parameters were not significantly correlated with their corresponding external loading at Waihai, indicating that there might be other factors influencing the water quality. (7) The contribution rates of internal release to the total loads of TN and TP at Waihai were estimated to be 7.6% and 8.9% respectively, suggesting that the reductions of both external and internal loading should be considered in order to significantly improve the water quality at Waihai of Lake Dianchi.

**Keywords:** Lake Dianchi; eutrophication; DYRESM; inflow volume; external loading

## 1. Introduction

A lake is a key node at the intersection of terrestrial ecosystem elements, playing roles in freshwater supply, flood storage and species conservation in the geosphere. Lakes are a valuable resource that human beings depend on to improve productivity through functions such as regulating runoff, developing irrigation and conducting shipping [1,2]. With the rapid growth of the global population and the gradual advancement of industrialization, urbanization and modern agriculture, a large amount of anthropogenic pollutants have been discharged into lakes, increasing the harm of eutrophication [3,4]. Lake eutrophication refers to a large increase in essential plant nutrients, such as nitrogen (*n*) and phosphorus (*p*),

in a lake, causing significant increases in the primary productivity of water ecosystems and resulting in the appearance of algal blooms, lower dissolved oxygen (DO) concentrations, reduced transparency, the death of aquatic animals, reduced biodiversity and damage to the normal habitat and function of the lake [5,6]. Studies [7,8] have found that eutrophication of a lake under natural conditions takes a long time to evolve, but disturbances from human activity can significantly accelerate the eutrophication process, shortening it from the original timeline of thousands of years to decades or even less. At the end of the 1990s, 61% of lakes worldwide were eutrophic [9], and the eutrophication rate of global inland waters had increased to 63% by 2012 [10]. The percentage of eutrophic lakes (reservoirs) among 110 important Chinese lakes (reservoirs) was 29% in 2020 [11]. The eutrophication of lakes has proven to be a threat to the sustainable development of human society.

Lake Dianchi is the largest freshwater lake on the Yunnan–Guizhou Plateau, playing key roles in the social and scientific development of Yunnan New Area [12]. However, it is in a moderate or severe eutrophic state all year round, and cyanobacterial blooms occur frequently, causing hidden dangers to the water environment and water safety of surrounding residents [13]. Guo et al. [14] found that $n$ and P loadings from urban sewage and agricultural runoff are the main sources of pollution in Lake Dianchi. Ma et al. [15] considered the hydrological characteristics of highland lakes, and they concluded that the long retention time of water bodies, weak exchange capacity and excessive nutrient loading have led to a faster rate of eutrophication in Lake Dianchi. Dong [16] found that soil erosion highly contributed to non-point source pollution in Lake Dianchi Basin and pointed out that rainfall, agricultural structure or rural population changes were conducive to increases in non-point source pollution loadings; because it could be concluded that external loading is the main root of eutrophication in Lake Dianchi, controlling external loading may be the first step to address eutrophication. As the links between a lake and terrestrial ecosystem in a basin, rivers are the key intermediate links for external loading control because land-based pollutants enter lakes by rivers, causing the deterioration of lake water and ecosystem quality [17]. About 70–80% of the annual water supplementation to Lake Dianchi comes from river inflow [18], and the average annual river total nitrogen (TN) and total phosphorus (TP) input can account for 80.2% and 78.8%, respectively, of the total external loading in Lake Dianchi [19], so accurate statistics regarding external loading by rivers is important for eutrophication management. However, there are more than 120 rivers flowing through complex paths into Lake Dianchi, resulting in poor statistics regarding water input and external loading [20]. Therefore, how to effectively invert the missing water volume and calculate external loading was the focus of this study.

A water balance equation is constructed by describing different hydrological processes in a basin as continuous water saving and flow processes, mainly using relevant factors, such as precipitation, temperature and runoff, as input in the water quantity inversion [21]. Zhang et al. [22] combined water deficit measurements (small ditches and rivers leading into the lake, farmland drainage entering the lake and groundwater infiltration and exfil­tration) into an uncertain incoming water term, and then they established a water balance equation for Lake Bosten with the incoming river flow, outgoing river flow and lake and evaporation consumption. Qin et al. [23] integrated incoming and outgoing flow, reservoir precipitation and evaporation and the loss of seepage from the reservoir to establish a water balance equation in Guanting Reservoir Station. Zan et al. [24] constructed a water balance equation for the Aral Sea based on regional rainfall, total evaporation and the amount of water entering and leaving the lake, and then they conducted a rough assessment of the total amount of groundwater data missing from monitoring. However, the authors of these articles mainly used historical data to build up simple mathematical equations, which are not computationally adequate for the dynamics of long-term time-series data. With the development of technology, researchers have effectively improved flood process forecasting accuracy by the machine learning method [25,26], and hydrological models have been widely applied to estimating variations of lake volume [27,28]. At present, the calculation principle of external loading is clear, mainly calculated through flow and water

quality concentration [29]. Therefore, the authors of this study used the computationally powerful hydrodynamic model DYRESM (Dynamic Reservoir Simulation Model) to calculate the elements of water balance in Lake Dianchi [30]. The DYRESM is a one-dimensional hydrodynamic model with the advantages of simple profile and convenient parameter-rate determination; it has been used in many domestic and international hydrological studies [31–33]. In this study, we first inverted the incoming water volume of Lake Dianchi from 2007 to 2019 and then calculated the external loading by combining the volume information with river water quality monitoring data in order to provide a scientific basis and reasonable suggestions for the management and ecological restoration of Lake Dianchi, as well as to provide reference methods for similar studies.

## 2. Materials and Methods

### 2.1. Study Site

Lake Dianchi is located in the southwestern part of the urban area of Kunming, the capital of Yunnan Province (Figure 1). It is one of the four major fractured tectonic lakes in China, and the only sewage-receiving body of the lakes in Kunming [34]. The Lake Dianchi Basin covers an area of 2920 km², accounting for about 0.75% of the land area of Yunnan Province but carrying nearly 23% of the province's gross domestic product (GDP) and 8% of the population [35]. With the development of urbanization and agriculture in Kunming, the nutrient loading of basin has significantly increased, resulting in the perennial deterioration of Lake Dianchi's water quality. The water area of Lake Dianchi is 309.5 km² (at an elevation of 1887.4 m), with a storage capacity of $1.56 \times 10^8$ m³ and an average depth of 5.3 m [36,37]. The southern part is called Waihai, which is the major part of Lake Dianchi, with a water area of 298.7 km² and average water resources that account for more than 90% of the total water resources of Lake Dianchi [37].

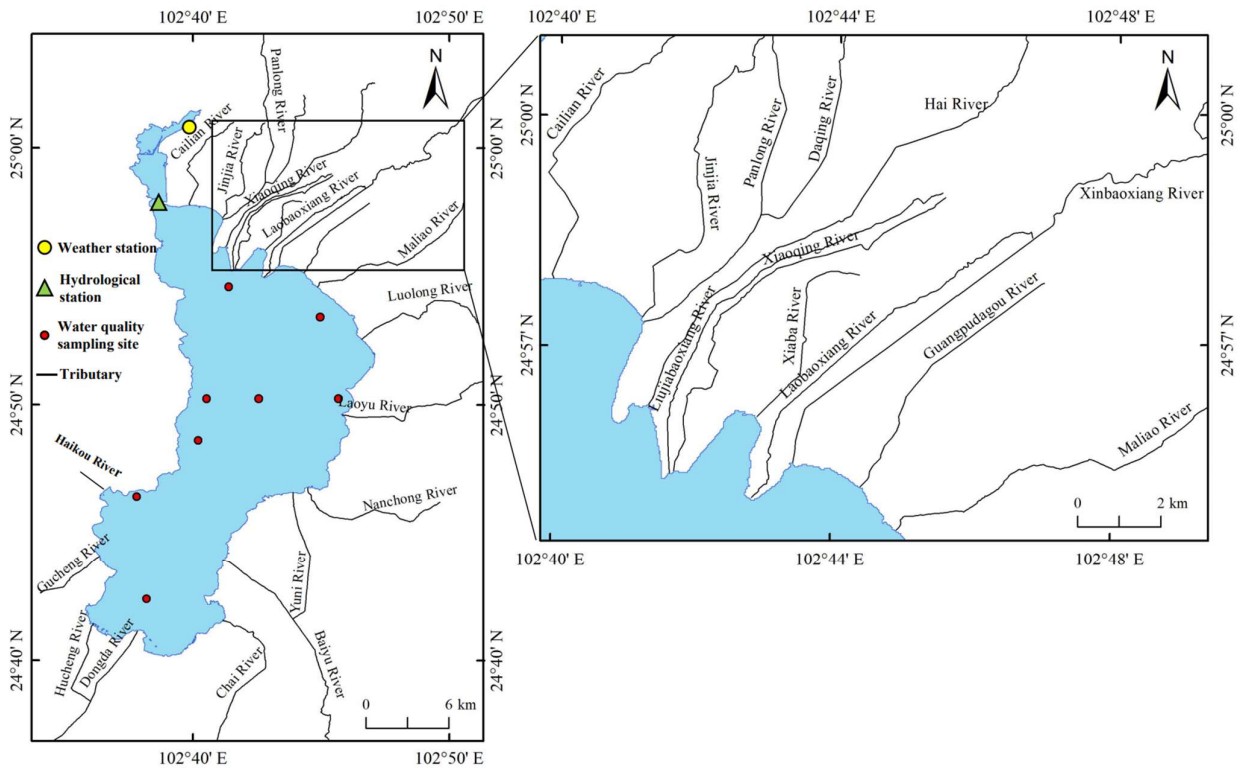

**Figure 1.** Distribution of meteorological stations, hydrological stations and main tributaries of Lake Dianchi.

Lake Dianchi Basin has a typical subtropical highland monsoon climate, with the mountains in the north blocking the northern cold streams in the winter, which allows the basin to have "four seasons like spring" all year round. The basin maintains a multi-year

average temperature between 14.6 °C and 15.9 °C and an annual temperature difference of about 12 °C. The lowest annual temperature occurs in January, and the highest annual temperature occurs in July [38]. The average multi-year rainfall is 986 mm, which can be divided into distinct dry and rainy seasons. The rainy season occurs from May to October each year, with rainfall accounting for more than 85% of the annual total, while the dry season occurs from November to next April, with rainfall accounting for only 15% or less of the annual total. The average multi-year evaporation is about 1871 mm, which is significantly higher than the average annual rainfall [39]. There are many rivers entering Lake Dianchi with a characteristic of "short flow near the source", and the special functions of transport, migration and sink determine the prominent position and role of rivers in the Lake Dianchi ecosystem [19]. In order to meet the ecological water demand, partial tailwater of Kunming urban sewage is discharged into Lake Dianchi after purification and treatment [40]. At present, there are artificially controlled outlets for Caohai and Waihai, which are the Xiyuan Tunnel at the northwest water area and the Zhongtan Gate at the southwest water area, respectively [41]. In addition, to support the urban development of Kunming and to meet the production and living water needs in the basin, Lake Dianchi's water resources are developed and utilized to 90%; furthermore, the total water supply in the basin was $8.20 \times 10^8$ m$^3$ in 2015, of which $1.36 \times 10^8$ m$^3$ was supplied by Lake Dianchi, indicating that water supply is also an important outflow pathway [42]. Since Waihai is the major body of Lake Dianchi, the annual volume of water in and out of the lake and the external loading are absolutely dominant in the total volume of Lake Dianchi [43], so the authors of this paper selected Waihai as the study area. The multi-year distance level changes in chlorophyll *a* (Chl−*a*), TN and TP concentrations in Waihai are shown in Figure 2, indicating that the water quality has significantly improved after years of treatment.

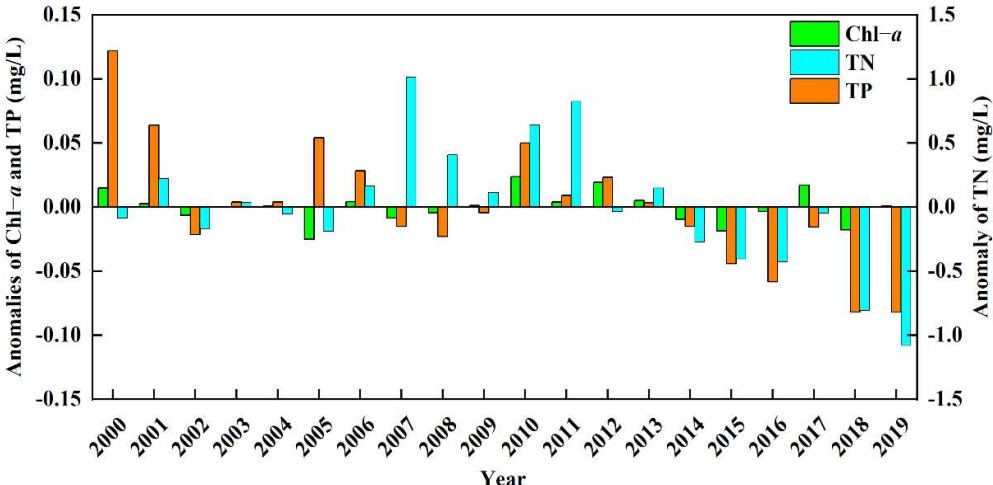

**Figure 2.** Annual anomalies of Chl−*a* (green), TN (blue) and TP (brown) at Waihai from 2000 to 2019.

### 2.2. Data Source

To ensure the accuracy of our results regarding water inversion and external loading, data of detailed hydrological, water quality, meteorological, topographical and river channels at Waihai were collected in this study. They are presented in Table 1 below.

Due to equipment failure and condition restriction, some monthly monitoring data of TN and TP in river channels were missing. When the monitoring data were missing for less than three months, a linear interpolation method was used to supplement. When the data were missing for more than three consecutive months, they were supplemented by calculating the monthly mean data of the previous and following years.

**Table 1.** Data information.

| Data Type | Data Period | Data Content | Data Source |
|---|---|---|---|
| Topography | \ | Water contour (elevation of Lake Dianchi bottom-area generation) | Kunming Dianchi & Plateau Lakes Institute |
| Meteorology | 2007–2019 | Daily data of weather station at Kunming | The China Meteorological Data Service Center |
| Inflow river | 2007–2019 | Monthly monitoring values of TN and TP | Dianchi Administration Bureau of Kunming |
| | 2007–2019 | Monthly monitoring values of river flow | |
| Tail water inflow | 2007–2020 | Yearly data of municipal treated sewage in Kunming; proportion of tail water discharged into Waihai after treatment in 2020 (31.08%) | Kunming Environmental Statement, Dianchi Administration Bureau of Kunming |
| Urban water supply | 2007–2019 | Yearly urban water supply of Kunming; in 2015, urban water supply accounted for 54.24% of whole basin, and Lake Dianchi water supply accounted for 16.59% of basin water supply | Kunming Statistical Yearbook [42] |
| Water regimen | 2007–2019 | Average daily water level of Waihai | Dianchi Administration Bureau of Kunming |
| Water quality | 2000–2019 | Monthly monitoring values of water quality at Waihai | Kunming Municipal Ecology and Environment Bureau |
| Water outflow | 2007–2019 | Daily measured flow of Haikou River | Dianchi Administration Bureau of Kunming |

There are 24 major input rivers around Waihai, and some of them presented a small amount of missing monthly water quality data that could be supplemented by using statistical methods. However, we found a large amount of missing data regarding instantaneous monthly river flow, which made it difficult to invert the water inflow volume. Therefore, the authors of this study collected measured data of river inflow, obtained the average values of historical flow for each river from January to December and then calculated the proportion of each river in the total annual flow after summing up the annual flow, which was used to allocate the inverse water volume. The percentage of missing measured data and each river flow in the total inflow volume from 2007 to 2019 are shown in Table 2.

**Table 2.** Information of rivers.

| River | Ratio of Missing Data (%) | Proportion of River Flow in Total Volume (Before 2012) | Proportion of River Flow in Total Volume (After 2012) | River | Ratio of Missing Data (%) | Proportion of River Flow in Total Volume (Before 2012) | Proportion of in River Flow Total Volume (After 2012) |
|---|---|---|---|---|---|---|---|
| Cailian River | 9.6 | 5.3 | 5.3 | Luolong River | 11.5 | 4.2 | 4.2 |
| Jinjia River | 59.6 | 8.6 | 8.6 | Laoyu River | 9.0 | 3.3 | 3.3 |
| Panlong River | 63.5 | 39.6 | 39.8 | Nanchong River | 19.9 | 0.6 | 0.6 |
| Daqing River | 12.2 | 11.2 | 11.3 | Yuni River | 24.4 | 1.4 | 1.4 |
| Hai River | 12.2 | 3.1 | 3.1 | Chai River | 15.4 | 1.5 | 1.5 |
| Liujiabaoxiang River | 33.3 | 0.4 | Cutoff | Baiyu River | 4.5 | 2.7 | 2.7 |
| Xiaoqing River | 72.4 | 1.1 | 1.1 | Cixiang River | 7.7 | 1.3 | 1.4 |
| Wujiabaoxiang River | 35.0 | 0.1 | Cutoff | Dongda River | 12.2 | 2.0 | 2.0 |
| Xiaba River | 62.2 | 1.8 | 1.9 | Hucheng River | 6.4 | 1.3 | 1.3 |
| Laobaoxiang River | 53.9 | 0.4 | 0.4 | Gucheng River | 1.9 | 0.4 | 0.4 |
| Xinbaoxiang River | 21.2 | 7.3 | 7.4 | Guangpudagou River | 23.1 | 0.9 | 0.9 |
| Maliao River | 20.5 | 0.9 | 0.9 | Yaoan River | 78.9 | 0.6 | 0.6 |

### 2.3. Model Description

DYRESM is a one-dimensional hydrodynamic model developed by the Centre for Water Research at the University of Western Australia that is mainly used for the simulation of lakes and reservoirs [30]. The model is capable of running alone to complete simulations of water temperature and salinity in the vertical direction of lakes and reservoirs, and it can be coupled with the CAEDYM (Computational Aquatic Ecosystem Dynamic Model) ecological model to simulate water quality and life processes of biological organisms in water areas, such as phytoplankton, fish and benthos, as well as the exchange of nutrients between water bodies and sediments [44].

The basic data required by DYRESM contained: (1) topographic basin data, such as the water surface area corresponding to different water depths that was calculated from the elevation–area relationship at the bottom of Lake Dianchi; we stratified the water body of Lake Dianchi at 0.1 m of water depth, with the maximum water depth being 11.5 m, and then separately calculated the water surface area at each depth. (2) The number of inflow channels, outflow channels and the elevation of the river entrance.

The DYRESM boundary conditions included: (1) meteorological files containing the daily data of solar short-wave radiation (W/m$^2$), air temperature (°C), water vapor pressure (hPa), average wind speed (m/s), cloudiness (0–1) or solar long-wave radiation (W/m$^2$), rainfall (m) and snowfall (m, set to 0 for areas without snowfall); (2) inflow and outflow files, with inflow files including daily inflow volume and water quality to the lake (m$^3$) and the outflow file mainly including daily outflow data (m$^3$).

The initial conditions of DYRESM were the water quality's distribution information in the vertical direction at the starting moment of simulation. The main physical parameters in the model and configuration files were debugged by drawing on the range of values provided in the literature for each parameter. The specific parameter values are shown in Table 3.

**Table 3.** Key parameters of DYRESM.

| Parameter | Value Range | Unit | Value in This Paper |
|---|---|---|---|
| Bulk aerodynamic momentum transport coefficient | $1.3 \times 10^{-3}$–$1.9 \times 10^{-3}$ [45,46] | \ | $1.3 \times 10^{-3}$ |
| Mean albedo of water | 0.07–0.084 [47,48] | \ | 0.075 |
| Emissivity of water surface | 0.94–0.96 [30,48] | \ | 0.96 |
| Critical wind speed | 3–6.5 [45,48] | m/s | 5.00 |
| Shear production efficiency | 0.06–0.084 [45,48] | \ | 0.08 |
| Potential energy mixing efficiency | 0.15–0.29 [48,49] | \ | 0.2 |
| Wind-stirring efficiency | 0.06–0.9 [32,50] | \ | 0.2 |
| Extinction coefficient | 0.2–0.8 [32,49] | m$^{-1}$ | 0.8 |
| Vertical mixing coefficient | 200–2500 [32,51] | \ | 200 |

### 2.4. Calculation Principle of Lake Volume Variation

The heat consumed by the evaporation of a lake surface is calculated according to following equation [52]:

$$Q_{lh} = min \left[ 0, \frac{0.622}{P} C_L \rho_A L_E U_a (e_a - e_s(T_s)) \Delta t \right] \quad (1)$$

where $Q_{lh}$ (quantity of latent heat) refers to the heat (J/m$^2$) consumed by the evaporation of the water surface during $\Delta t$, $P$ is the atmospheric pressure (hPa), $C_L$ is the latent heat conduction coefficient ($1.3 \times 10^{-3}$) of wind speed at a 10 m reference height, $\rho_A$ is the air density (kg/m$^3$), $L_E$ ($2.453 \times 10^6$ J/kg) is the latent heat of water evaporation [47], $U_a$ is the wind speed at a 10 m reference altitude (m/s), $e_a$ is the vapor pressure of air (hPa), $e_s$ (saturation vapor pressure) is the saturated vapor pressure (hPa) under the condition of water surface temperature ($T_s$) and $\Delta t$ is the calculated time step of model, which is set to 3600 s [52].

The formula for calculating the mass change (kg) of the $N$th-layer in a lake caused by evaporation is as follows [52]:

$$\Delta M_N^{(lh)} = \frac{-Q_{lh} A_N}{L_E} \tag{2}$$

where $\Delta M_N^{(lh)}$ represents the mass changes of water (kg) caused by evaporation in the $N$th-layer ($N \geq 1$, $N = 1$ means the surface layer of the water column), $A_N$ is the surface area of $N$th-layer and other variables are as mentioned above.

The calculation formula of water level rise (m) caused by precipitation is as follows [52]:

$$r_h = R_h \frac{\Delta t}{N_d} \tag{3}$$

where $r_h$ is the water level changes (m) of the $N$th-layer caused by precipitation, $R_h$ is the total daily rainfall (m) and $N_d$ is the duration of daily rainfall (s).

The calculation formula of water mass change (kg) in different layers by precipitation is as follows [52]:

$$\Delta M_N^{(rain)} = \rho_N A_N r_h \tag{4}$$

where $\Delta M_N^{(rain)}$ is the mass changes (kg) caused by precipitation of the $N$th-layer, $\rho_N$ is the water density (kg/m$^3$) and other variables are as mentioned above.

The formula for calculating the total water mass change in the $N$th-layer of the lake caused by evaporation and precipitation is as follows [52]:

$$\Delta M_N = \Delta M_N^{(lh)} + \Delta M_N^{(rain)} \tag{5}$$

According to the five above-described formulas, DYRESM can automatically calculate the daily evaporation of a lake surface and the corresponding water level variation.

### 2.5. Calculation Principle of Water Compensation Method

The principle of our water compensation method is shown in Figure 3. After configuring the original inflow and outflow files of DYRESM, considering the influence of lake precipitation and evaporation on the storage capacity, the daily simulated water level was obtained by model using the area and volume data corresponding to different depths provided in the underwater topographic map (scale 1:2000). The water level storage capacity curve of Lake Dianchi was constructed by linear fitting. The used fitting equation was y = 2.89x − 5435.77, where y is storage capacity ($\times 10^8$ m$^3$), x is water level (m) and correlation coefficient (r) = 0.99. Based on the water level–storage capacity curve, the model was able to calculate the daily simulated storage capacity and measured storage capacity, respectively. Then, the difference between the storage capacity of the next day and that of the previous day was calculated, allowing us to obtain daily simulated storage capacity difference and the measured storage capacity difference data.

The daily compensation value was obtained in the model by subtracting the difference between daily measured storage capacity difference and simulated storage capacity difference. If the compensation value was positive, simulated storage capacity was lower than measured storage capacity, meaning that inflow volume in the model needed to be increased; we set a virtual river channel in the inflow file to supplement increased inflow into the virtual river channel. A negative value indicated that the outflow of model needed to be increased. Taking the absolute value of the compensation value and adding it to the outflow flow to complete the primary water volume compensation calculation, this was followed by a comparison of the simulated water level results with the measured water level. If the error could be ignored, the calculation was stopped. If the error was obvious, water compensation began again. Then, the new inflow and outflow compensation values were calculated so that the inflow (outflow) could be accordingly modified. Following water compensation, the daily water volume of the virtual river was close enough to the daily total inflow of real rivers to be used in subsequent work.

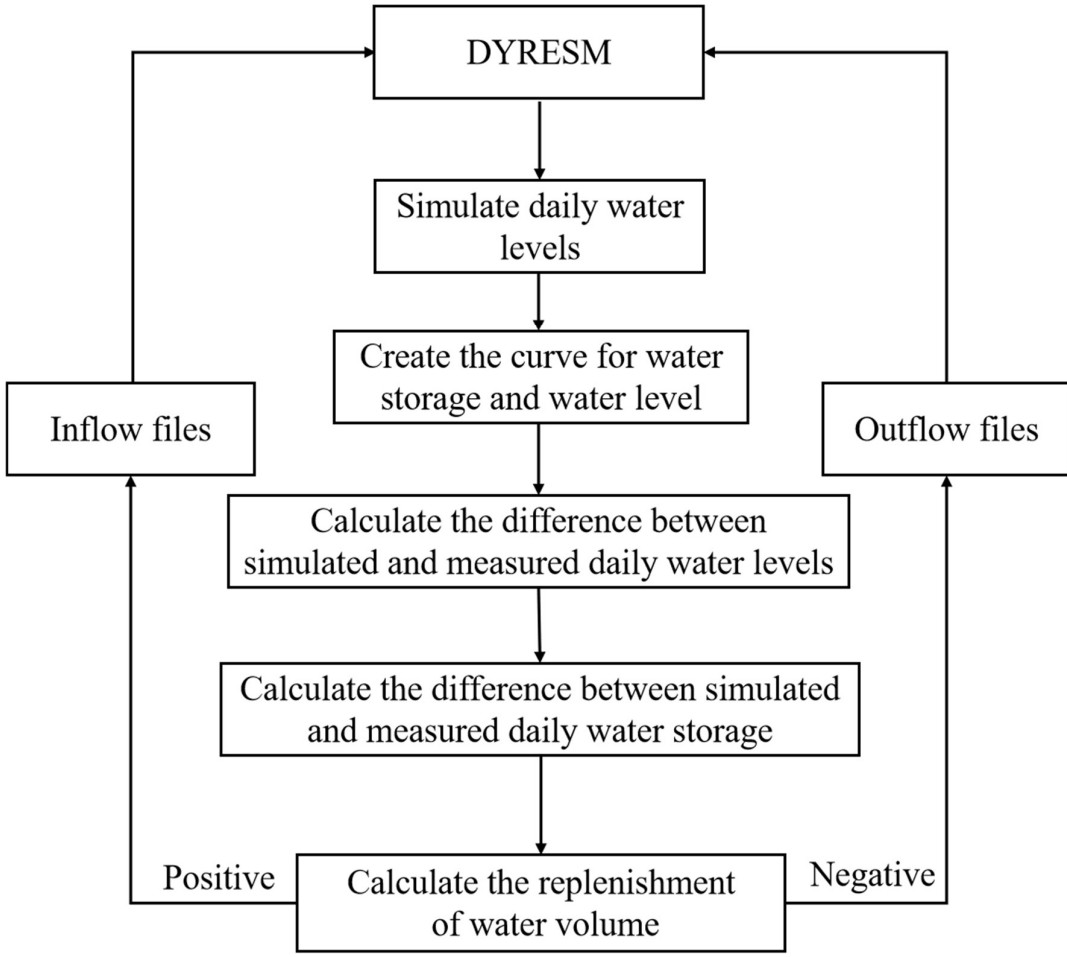

**Figure 3.** Flowchart of water compensation method for DYRESM.

### 2.6. Original Inflow and Outflow Files of DYRESM

According to Table 2, river inflow data were seriously missing and could not be configured for the model inflow file. Statistics regarding urban sewage treatment capacity over the years were obtained from the Kunming Environmental Statement. The annual tail water inflow could be calculated based on the proportion of tailwater flow into Waihai after sewage treatment in 2020. The tail water inflow was distributed every day to obtain original inflow profiles containing the tail water data. The original outflow document included the daily measured discharge of Haikou River and the daily water supply of Lake Dianchi. The calculation steps of daily water supply were as follows: First, the urban water supply of Kunming over the years was counted. Second, according to the proportion of urban water supply in Lake Dianchi Basin in 2015, the water supply of basin over the years was obtained. Finally, based on the proportion of Lake Dianchi water supply in the basin water supply in 2015, the water supply of Lake Dianchi over the years was calculated, and the daily water supply in the year was found to be equally distributed. At the same time, since only the water level change was considered, the water quality concentration in and out of the lake was set to 0.

### 2.7. Evaluation Standard of Model Error

The model error was verified by calculating the root mean square error (*RMSE*) between the measured and model-simulated values, and the Nash efficiency coefficient

(*NSE*) and correlation coefficient (*r*) between measured and simulation values [48]. *RMSE* and *NSE* were calculated as follows:

$$RMSE = \sqrt{\frac{1}{N}\sum_{i=1}^{N}(S_i - O_i)^2} \tag{6}$$

$$NSE = 1 - \frac{\sum_{i=1}^{N}(S_i - O_i)^2}{\sum_{i=1}^{N}(O_i - \overline{O})^2} \tag{7}$$

where $O_i$ is the measured value, $S_i$ is the simulated value, $\overline{O}$ refers to the arithmetic average of measurements and $N$ is the number of data. *RMSE* results can explain the dispersion degree of samples; the smaller the value, the better the simulation effect. The *NSE* is a dimensionless statistical parameter that is commonly used to describe the fitting accuracy of models ($NSE \leq 1$); $NSE = 1$ indicates a complete fit, and $NSE \leq 0$ indicates that the fitting degree is very poor. When the *NSE* is positive, the simulated value can better express the law of the measured value than the average of the measured value. The closer the *NSE* value is to 1, the better the fitting degree and the better the simulation effect.

### 2.8. Calculation Method of External Loading by River

The dry and rainy seasons are distinct in Lake Dianchi Basin, and the discharge into Waihai of each month significantly varies. Therefore, we allocated the yearly retrieved water volume according to the annual inflow proportion of each river, distributing the annual water volume of each river to the month according to the proportion of historical monthly inflow. We used the monthly measured values of TN and TP of each river as the monthly concentration to obtain the external loading of each river channel, and the total external loading input by river channel was obtained by adding external loading.

## 3. Results

### 3.1. Waihai Water Level Simulation

We implemented a water balance analysis from January 2007 to December 2019. After this calculation, the simulated water level clearly agreed well with the observed water level (Figure 4). Before the calculation, the simulated water level at Waihai continued to decline because the annual evaporation in Lake Dianchi Basin is greater than its precipitation [39]. The DYRESM accurately reproduced the water level, with a high coefficient of determination and small relative error values (*RMSE* = 0.0072 m; *NSE* = 0.99; *r* = 0.99). The maximum measured water level was 1887.56 m on 11 August 2015, and the simulated water level on that day was 1887.57 m. The lowest water level occurred on 24 May 2010 (1886.35 m), and the simulated water level was also 1886.35 m. This showed that the DYRESM could reflect fine variations and extreme conditions in measured water levels well after calculation. From 2009 to 2010, the water level at Waihai significantly decreased, which was completely different from other periods and probably because of the drought in Yunnan Province in 2009 [53].

### 3.2. Retrieval Results of Water Inflow

We calculated the total annual inflow by river (Figure 5a). River flow is closely related to rainfall in the basin [54], so the annual total lake inflow was fitted with the annual total rainfall to verify the accuracy of the calculation results, which are shown in Figure 5b. The correlation coefficient between annual total runoff and annual total rainfall was 0.74 (Figure 5b), indicating a significant relationship between inversion water volume and precipitation, consequently demonstrating that the DYRESM's inversion water amount was feasible. From 2007 to 2019, the annual total lake inflow by river was consistent with the changing trend of the annual total precipitation (Figure 5a). During the study period, the annual average inflow volume of Waihai was about $6.69 \times 10^8$ m$^3$, which was consistent with the annual average land water inflow of $6.97 \times 10^8$ m$^3$ of Lake Dianchi [55]. The

inflow volume significantly decreased after 2009, and it reached the lowest level of only $3.24 \times 10^8$ m$^3$ in 2011. According to the Kunming Statistical Yearbook, 2011 was the third consecutive year of drought relief in Kunming, with a total precipitation of 697.80 mm, 29 cut-off river channels and an accordingly decreased inflow volume. In 2017, the lake inflow was as high as $10.16 \times 10^8$ m$^3$, and the total annual precipitation was 1186.4 mm, both of which were the highest values from 2007 to 2019.

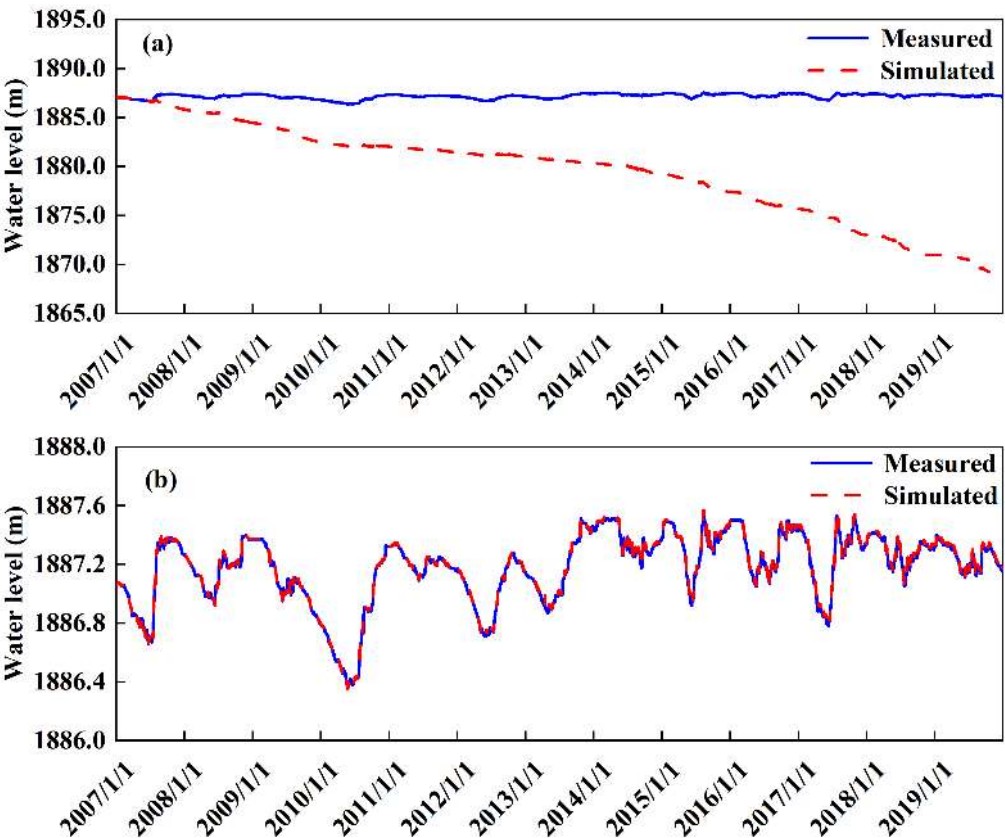

**Figure 4.** Simulated and measured daily water levels from 2007 to 2019: (**a**) before water compensating; (**b**) after water compensating.

### 3.3. Variation within the Year of Water Inflow

The simulated and calculated changes in inflow and precipitation with the year showed a single-peak trend of first rising and then decreasing. The water inflow was as high as $1.15 \times 10^8$ m$^3$ in July, and the average precipitation was 212 mm, both of which were first within the year. In the following August, the average inflow volume and precipitation were $1.01 \times 10^8$ m$^3$ and 200.44 mm, respectively. These results are consistent with the viewpoint summarized by Chen that "Flood season in Lake Dianchi Basin is mainly concentrated in July and August" [56]. Through fitting calculation, it was found that there was a close relationship between retrieval inflow and precipitation in the year (Figure 6b; $r = 0.98$), which proves the important significance of precipitation forecast in the flood control and waterlogging work of Lake Dianchi.

### 3.4. Calculation Results of External Loading by Riverway

To verify the accuracy of the river external loading calculated by simulated volume and measured water quality, we collected TN and TP loading data from inflow rivers during the study period and analyzed them with the calculated loading. The specific data are shown in the following table.

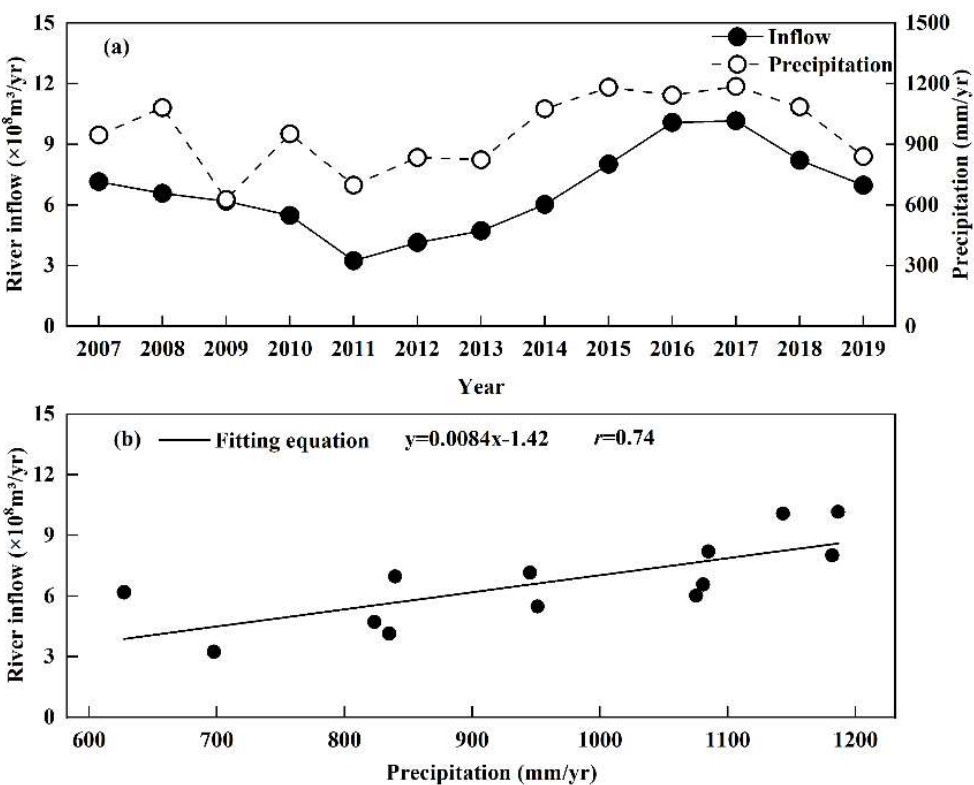

**Figure 5.** Yearly variations of total river inflow and precipitation (**a**) and their fitting relationship (**b**) for 2007–2019.

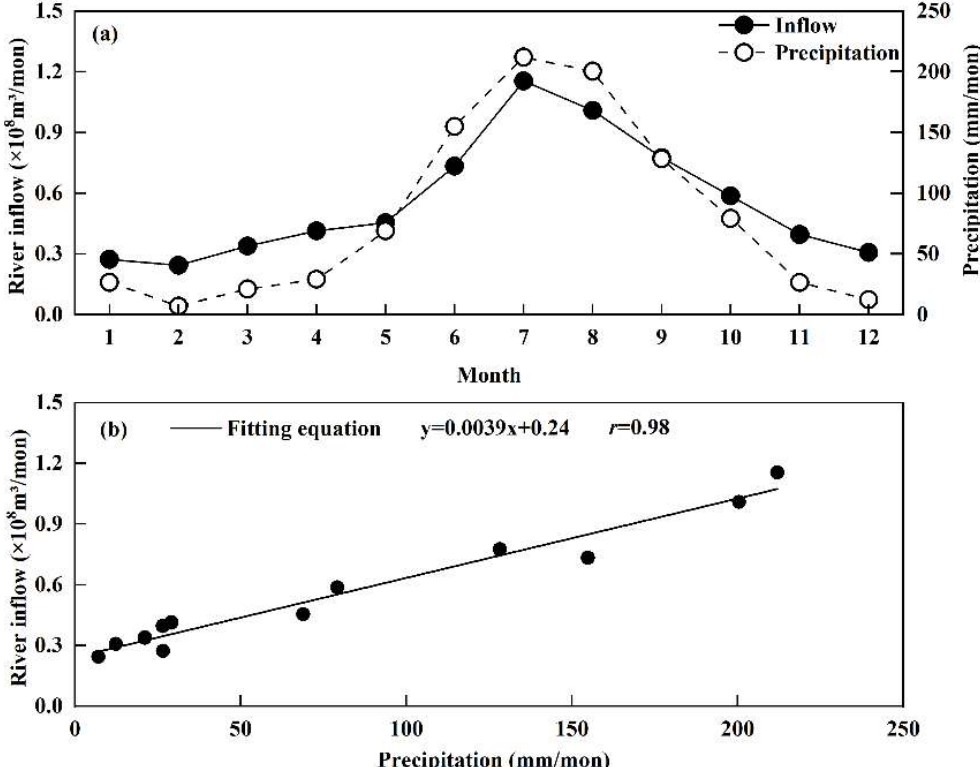

**Figure 6.** Intra-annual variations of total river inflow and precipitation (**a**) and their fitting relationship (**b**) for 2007–2019.

In some publications, only the basin emissions of TN and TP or the total amount of external loading of Lake Dianchi (including Caohai) have been studied. Therefore, in this study, we used the calculation coefficients of TN loading (the total external loading of Lake Dianchi accounted for 64% of the whole basin and the total amount of Waihai accounted for 73% of the total external loading of Lake Dianchi) and TP loading (the total amount of external loading of Lake Dianchi accounted for 60% of the whole basin and the total amount of Waihai accounted for 90% of the total external loading of Lake Dianchi) according to the specific values given in the "The 14th Five-Year Plan period for water environmental protection and governance in Lake Dianchi Basin". Additionally, the abovementioned coefficients were used to calculate the total amounts of TN and TP in Waihai.

From the data in Table 4, it can be seen that the calculation amount of river loading before 2011 was often higher than the actual amount. This was due to the fact that the loading calculation coefficients were based on Waihai data in 2009, resulting in a reduction effect in external loading before "The 12th Five-Year Plan period" being higher than in reality. However, the change trend of river loading and the total amount of Waihai remained roughly the same. During the study period, the average annual TN loading input by the river channel was 5480 t, and the average annual input loading of TP was 295 t. In 2011, due to a drought in the basin, the amount of water entering Waihai was significantly reduced, resulting in input loadings of TN and TP by the river channel of 2616 t and 107 t, respectively, which were minimum values in the calendar year. From the perspective of time scale, the river loadings of TN and TP declined year by year: the TN loading into Waihai in 2007 was 9080 t and the loading fell to 3728 t in 2019, with a reduction rate of 59%. The TP loadings into Waihai in 2007 and 2019 were 713 t and 115 t, respectively, and the reduction rate was as high as 84%.

**Table 4.** Annual external loading of Waihai from 2007 to 2019.

| Year | TN Loading by Riverway (Calculated Value, Ton) | Total External Loading of TN (Literature Value, Ton) | TP Loading by Riverway (Calculated Value, Ton) | Total External Loading of TP (Literature Value, Ton) | Data Source |
|------|------|------|------|------|------|
| 2007 | 9080 | 7452 | 713 | 782 | [20] |
| 2008 | 8290 | 4990 | 772 | 294 | [57] |
| 2009 | 8782 | 6231 | 512 | 697 | [58] |
| 2010 | 6265 | 4268 | 284 | 390 | [59] |
| 2011 | 2616 | \ | 107 | \ | \ |
| 2012 | 3948 | 4299 | 161 | 370 | [60] |
| 2013 | 4145 | 3978 | 197 | 331 | [61] |
| 2014 | 4167 | 5358 | 207 | 538 | [62] |
| 2015 | 4235 | 5656 | 179 | 495 | Lake Dianchi Protection and Governance Plan (2016–2020) |
| 2016 | 5602 | 6590 | 203 | 566 | Lake Dianchi Protection and Governance Three-year Tackling Action Implementation Plan (2018–2020) |
| 2017 | 5906 | 3842 | 205 | 390 | Lake Dianchi Protection Plan (2018–2035) |
| 2018 | 4472 | 5109 | 183 | 450 | [63] |
| 2019 | 3728 | 3884 | 115 | 397 | The 14th Five-Year Plan for Water Environment Protection and Management of Lake Dianchi Basin (2021–2025) |

The TN and TP loadings of rivers showed regular changes with the year (Figure 7). TN loading by rivers reached high values of 1035 t and 687 t in July and August, respectively, accounting for 19% and 13% of the whole year's loading, which were similar to the proportions of total inflow volume in July and August. The TN loadings by rivers in the rainy season (May–October) and the dry season were 3473 t and 2007 t, respectively, accounting for 63% and 37% of the annual loading. The TP loadings in the rainy and dry seasons were 197 t and 98 t, respectively, accounting for 67% and 33% of the annual loading. These results show that the river loading has distinct annual distribution characteristics. In the rainy season, with the significant increase in river inflow, the external loading of the river significantly increases. Therefore, different measures should be taken to control the loading of river in the rainy and dry seasons.

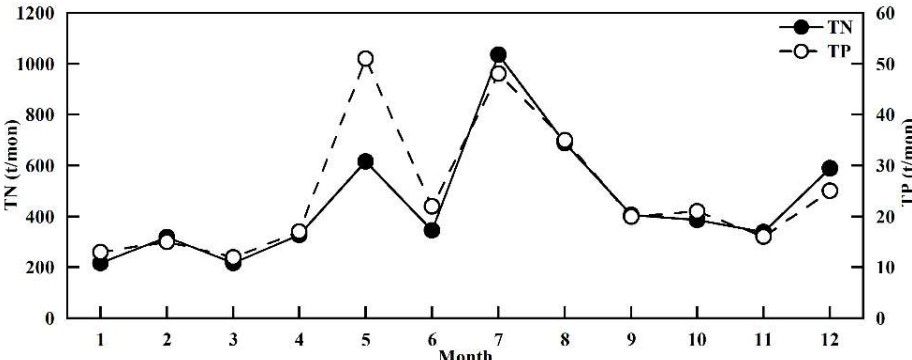

**Figure 7.** Intra-annual variation of TN and TP loadings for inflow rivers.

In order to clarify the focus of river management, the authors of this study calculated the multi-year proportion of 24 rivers' loading in the total pollutants and highlighted the main external input channels (Figure 8). Panlong River, Daqing River, Jinjia River, Xinbaoxiang River, Cailian River and Haihe River were found to account for more than 5% of the total input of TN and TP at Waihai. Panlong River was found to be the most important source of external loading, with TN and TP loadings accounting for 32.7% and 23.8%, respectively, due to the fact that the amount of water entering Panlong River is about $2.50 \times 10^8$ m$^3$ and the abundant water volume provides convenient conditions for receiving the basin's pollutants [64]. The proportions of the total TN and TP loading by river in Daqing River were second only to Panlong River at 18.1% and 20.4%, respectively, but the inflow volume of Daqing River was found to account for only 11.3%, indicating that the water quality of Daqing River is poor. This is because the upstream tributary called Mingtong River belongs to the sewage channel and the terminal sewage interception gate has the risk of overturning the weir in the rainy season, thus causing Daqing River to face risks of deteriorating water quality. Haihe River was shown to be similar to Daqing River, with an inflow rate of only 3.10%, but it was found to carry 6.1% of TN loading and 11.2% of TP loading, indicating that the water quality of Haihe River is worse than that of Daqing River. This water quality issue is possibly due to the incomplete diversion of rainwater and sewage in the drainage system of river basins, which allows the domestic sewage of villages to easily overflow into the river during the rainy season.

### 3.5. River Loading and Water Quality

From Figure 2, it can be seen that the water quality of Waihai was significantly improved after treatment. Compared to 2007, the improvement rates of TN and TP of Waihai in 2019 were 68.8% and 50%, so the pollution of the water body was mitigated. There was no obvious trend in the change in Chl−*a* concentration, which indicated that there are differences in the influencing factors of Chl−*a* and other water quality indicators and that more targeted treatment measures are needed. A Pearson correlation analysis of yearly data between river loading and Waihai water quality from 2007 to 2019 was performed, and the results are shown in Table 5.

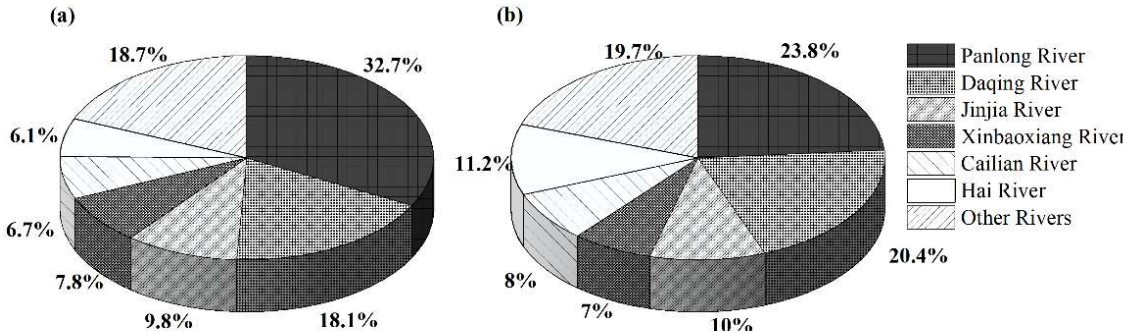

**Figure 8.** Proportion of external loadings of TN (**a**) and TP (**b**) in the main inflow rivers.

**Table 5.** Pearson's correlation between river loadings and water quality parameters from 2007 to 2019.

| Indicators | Chl−*a* (mg/L) | TN (mg/L) | TP (mg/L) | *n* |
|---|---|---|---|---|
| TN loading (t) | 0.008 | 0.44 | 0.13 | 13 |
| TP loading (t) | −0.13 | 0.54 | 0.12 | 13 |

This table shows that there was no significant correlation between external loading by river and water quality index of Waihai, indicating that there are other influencing factors of water quality besides river loading, implying that the influence of loading input by river is not yet possible without deeper research.

### 3.6. Effects of Different Pollution Sources

Lake pollution is divided into two types (internal source and external source), and external inputs are dominated by river loading, although atmospheric deposition also has a significant impact on lake pollution that is more significant in highland lakes [65]. The tailwater discharged after sewage treatment also carries certain pollutants. In this study, TN and TP data of Waihai regarding different pollution sources in 2014 were compiled based on the literature, and the results are shown in Table 6.

**Table 6.** TN and TP loadings from different sources at Waihai in 2014.

| Data Type | TN Loading (t) | TP Loading (t) | Data Source |
|---|---|---|---|
| River loading | 4167 | 207 | Calculation result |
| Tail water loading | 1953.18 | 65.11 | Calculated by tail water inflow and water quality mission standards |
| Atmospheric deposition | 407.73 | 34.51 | [66] |
| Internal pollution | 539.84 | 29.88 | Basic investigation report on total volume control at Dianchi Basin |

River loading was found to account for 59% and 61.5% of the total TN and TP, respectively, due to the significantly higher population density in Lake Dianchi Basin (which is close to twice that of Lake Chaohu) that has enabled the river loading and pollution absorption pressure in Lake Dianchi to become more prominent [67]. Over the years, in order to collect and treat point source sewage and surface source sewage, Kunming has vigorously promoted the construction of urban domestic sewage treatment plants, but there is still a large gap between sewage discharge standards and surface water environmental quality standards that has resulted in a large tailwater loading. Lake Dianchi Basin is also an important flower and vegetable production base, with fertilizer use nearly 2.5 times higher than the national level [68]. Because of low rainfall in the dry season, *n* and *p* particulates from biomass burning, industrial production emissions and fertilizer application losses are enriched over Lake Dianchi; in the rainy season, they enter the lake with precipitation

and increase the lake pollution [66]. Lake characteristics largely determine the nutrient change pattern of Lake Dianchi, and dynamic disturbances and wind and wave processes in shallow lakes are likely to cause sediment suspension and internal pollution release [69], so the exchange of internal pollutants at the water–sediment interface will accelerate water quality deterioration or seriously affect nutrient loading reduction [70].

## 4. Discussion

### 4.1. Analysis of Water Quantity Retrieval and External Loading Results

The inversion of river volume by the DYRESM was a lake water balance model based on the core principle that the increase in lake water over a certain period is equal to all the water entering the lake minus all the water discharging from the lake, so the inversion data at Waihai also contained some groundwater-dominated uncertainty. Groundwater is often regarded as an important recharge source [22], but there are no major rivers transiting Kunming, and the regional water recharge mainly relies on seasonal rainfall, with a groundwater resource of about $1.98 \times 10^8$ m$^3$ [71]. With the development of society, groundwater levels in seven water-rich blocks in the Kunming area continued to decline from 2004 to 2013, accompanied by water level decreases ranging from 0.2 m to 12.6 m [72], so the groundwater recharge to Lake Dianchi could be ignored in this study. The fit verification between river inflow and Kunming precipitation on monthly or yearly scales showed that there was a strong connection between inverse inflow and precipitation, thus proving that precipitation is the fundamental water source of Lake Dianchi and providing a basis for flood prevention through precipitation forecasting. This analysis shows that the results of total inflow by the DYRESM inversion were reliable, and it is reasonable to consider all the total inflow as the river flow in Lake Dianchi Basin.

The measured flow data of river channels were seriously missing (Table 2), meaning the vacant values could not be replaced with statistical methods. Therefore, the authors of this study calculated the annual flow and monthly flow percentage of each river channel to allocate the inverse water volume. Although this method has errors, the real river flow should have distinctive monthly characteristics because precipitation is the main recharge source of the river channel (Figure 6), so the allocation method of this study reflected the monthly changes, and the error value was reduced. In addition, the authors of this paper used the monthly monitoring values of river water quality to represent the daily water quality in each month, which may have led to errors in external loading results. Because the main inflow rivers are located in the northeastern shore and pass through the main human activity area, the basin's pollutants tend to sink into rivers in the rainy season with short-term heavy rainfall, causing the temporary elevation of TN and TP in rivers. Therefore, the accuracy of using river water quality in extreme weather to represent the prevailing conditions in those months was limited. In summary, the frequency of water quality monitoring should be increased to solve this problem in the future.

### 4.2. Analysis of Water Quantity Retrieval and External Loading Results

Due to the difficulty of "Three rivers and Three lakes" in the key national governance, the governance process of Lake Dianchi is significant. After years of investment, the pollution degree of Lake Dianchi has been effectively alleviated. From 1993 to 2015, the water quality of Lake Dianchi was always deteriorating in the inferior V class, but the water quality changed to V class in 2016 and remained stable in IV class in 2018–2019 [43]. These changes demonstrate the gradual emergence of the treatment effect, a result that is consistent with the trends of water quality and river loading in Figure 2 and Table 3. Although TN and TP loadings declined from 2007 to 2010, they remained high. During the period of the "11th Five-Year Plan", Kunming began conducting engineering governance of inflow rivers, but the river situated at the north shore of Lake Dianchi flows through a main urban area with a large amount of urban sewage and rainwater pollution [18], resulting in limited reduction in river loading [18]. Due to the drought situation, the inflow volume was low in 2011, so external loading in that year was low. In addition, the local

government implemented regulations on river management of Kunming in 2010, ensuring the effect of comprehensive regulation in the system while also increasing the investment in river treatment, which laid a solid economic foundation for comprehensive regulation [18]. In conclusion, effective lake environmental management requires long-term system and economic support. During the "12th Five-Year Plan period", Lake Dianchi Basin water pollution control and eutrophication comprehensive control technology were included in a special water project oriented to the whole basin. Accordingly, comprehensive control was enacted and industrial point source pollution was controlled. Attention to inflow river and internal pollution treatment have kept increasing since then. Six major projects, including pollution interception around the lake, agricultural and rural non-point source treatment and ecological restoration construction, have been fully implemented, and the pollution loading into Lake Dianchi has been significantly reduced [61]. During the "13th Five-Year Plan period", the water quality of Lake Dianchi improved as a whole, and cyanobacteria blooms have continued to improve. However, due to temporal and spatial instability, non-point source pollution, internal source release and soil erosion have replaced point source pollution and become the main loading sources [43]. Therefore, river loading has remained at a low level after 2016, but the improvement effect of lake water quality has shown partial hysteresis.

### 4.3. Influence of Internal Pollution on Waihai Water Quality and Control Measures

Pearson's test results showed that there are other factors besides river loading affecting water quality at Waihai. According to Table 6, internal pollution was found to account for 7.6% and 8.9% of the total TN and TP loadings, respectively, suggesting that the role of internal pollution on the water quality of Lake Dianchi should not be ignored. Sediment nutrients can enter shallow lakes through not only molecular diffusion or concentration gradient diffusion (static diffusion) but also sediment resuspension and changes in conditions at the water–sediment interface; turbulent diffusion causes much higher internal release than static diffusion [73]. Zhu et al. concluded that a wind speed of above 8 m/s may cause a large amount of suspension of sediments in Lake Taihu, and the concentration of dissolved TP may increase up to 100% during strong winds [74]. Luo et al. used field investigations combined with data and mathematical interpolation methods to calculate that, when the wind speed reached 20 m/s, it could result in the suspension of about $2.75 \times 10^8$ m$^3$ of sediment in the upper 30 cm of Lake Taihu [75]. Zhang simulated water body changes in the middle of Lake Chaohu during the sediment resuspension period through laboratory experiments and concluded that the different intensities and durations of external disturbance directly affected the suspended state of sediment particles [76]. Lake Dianchi is a shallow lake with a low water-exchange rate, and a large number of pollutants are deposited at the bottom. When external loading is controlled, sediments in the lake will continue to affect the water quality [77]. In 2012, 6800 t of TN from the basin's non-point source was loaded into Lake Dianchi, and sediment $n$ comprised nearly 67.5% of non-point pollution loading, showing that sediments of internal pollution are very serious [78], as well as leading to external loading reduction benefits that can only be offset to some extent with truncated external loading measures to reduce the trophic level of Lake Dianchi in a short time. The bottom mud of Lake Dianchi contains a variety of humus and organic matter, so Kunming enacted the measures of "environmental protection dredging" and complete reduction through harmless resource treatment of the bottom mud. By 2018, Kunming had cleared $15.17 \times 10^6$ m$^3$ of sediment and used the dredged sediment for ecological basement restoration and ecological forest construction in the low-land area around Lake Dianchi [60,79].

Over the years, local government and civil society have invested huge amounts of resources into river management and ecological restoration, but the restoration of Lake Dianchi has always been a long-term and systematic process. In this study, the amount of inflow volume entering Waihai was analyzed through model inversion. There are many tiny ditches around Lake Dianchi that are not included in the monitoring range, and

the accumulation of pollutants in these ditches represents external loadings that cannot be ignored. Therefore, in the next stage of research, a circumnavigation survey of Lake Dianchi will be attempted, small intakes, including ditches, will be counted and water quality monitoring will be regularly carried out to obtain more accurate data regarding inflow volume and external loading.

## 5. Conclusions

1.  The DYRESM can effectively capture extreme changes in water levels with an *RMSE* value of 0.0072 m between simulated and measured water levels and an *NSE* as high as 0.99.
2.  During the period of 2007–2019, the multi-year average annual water inflow to Waihai was about $6.69 \times 10^8$ m$^3$, and there is a good fit between water inflow and precipitation in Kunming on an annual scale ($r = 0.74$), with a higher fitting coefficient between intra-annual inflow and precipitation ($r = 0.98$).
3.  The external loading by rivers has decreased year by year, although river loading remained at a high level from 2007 to 2010. In 2011, the TN loading dropped to 2616 t and the TP loading dropped to 107 t due to a drought in the basin, and the river loading in subsequent years basically remained at a low level.
4.  River loading was found to have clear intra-annual variation characteristics, and the contributions of TN and TP river loadings in the rainy season were 63% and 67% of the annual amount, respectively, indicating that river management should focus more on loading reduction in the rainy season.
5.  Panlong River, Daqing River, Jinjia River, Xinbaoxiang River, Cailian River and Hai River are the focuses of treatment, and the sum of the loading of these rivers was found to account for 81.3% (TN) and 80.3% (TP) of the total river input.
6.  Pearson's analysis results showed that there was no significant correlation between annual external loading and Waihai water quality, indicating the existence of other factors that influence water quality besides source input.
7.  The contribution rates of internal pollution to the total amount of TN and TP were found to be 7.6% and 8.9%, respectively, indicating that the internal control of Lake Dianchi should not be ignored.

**Author Contributions:** Conceptualization, R.Z., L.L. and H.L.; methodology, L.L.; software, L.L.; validation, J.L., F.G., G.W., L.C. and J.Z.; investigation, M.P., F.H., C.L., D.M. and H.L.; resources, L.L.; data curation, L.L., M.P., F.H., C.L. and H.L.; writing-original draft preparation, R.Z., L.L., J.L., F.G., G.W., L.C., J.Z. and T.S.; writing-review and editing, R.Z., J.L., F.G., G.W., L.C., J.Z. and T.S.; supervision, L.L., M.P. and H.L.; project administration, L.L., M.P., F.H. and H.L.; funding acquisition, L.L. All authors have read and agreed to the published version of the manuscript.

**Funding:** This research was funded by Yunnan Provincial Department of Science and Technology (202001BB050078), Yunnan University (C176220100043), National Natural Science Foundation of China (41671205) and Dianchi Administration Bureau of Kunming (K207002003021).

**Institutional Review Board Statement:** Not applicable.

**Informed Consent Statement:** Not applicable.

**Data Availability Statement:** The data that support the findings of this study are available from the corresponding author upon reasonable request.

**Acknowledgments:** The authors would like to thank the English editors from MDPI for their language editing on the manuscript.

**Conflicts of Interest:** The authors declare no conflict of interest.

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
