# Peer review of "Estimations of Water Volume and External Loading Based on DYRESM Hydrodynamic Model at Lake Dianchi"

_water, doi:10.3390/w14182832_

Round 1

Reviewer 1 Report

The authors used the DYRESM model to estimate the water volume entering Waihai of Lake Dianchi for 2007—2019 without historical observation data, and further estimated external loading for Waihai based on monthly water quality monitoring data for the inflow rivers. The calculated inflow volume and external loading were statistically analyzed and compared to literature data to verify their accuracy. The model approach used in this paper is very useful for other similar research. The results of external loading estimation can provide substantial information for water environmental management for Lake Dianchi. I would recommend the paper is accepted after minor revision following the comments below.

(1) Add the latest references about estimations of inflow volume and external loading for lakes or reserviors.

(2) I wonder if the ground water is an important water supply or not for Lake Dianchi. Please list the proporations of ground water, riverine inflows and rainfall as well as evaporation if possible.

(3) The TN and TP loadings were high from 2007 to 2010, but then declined rapidly and remained at a low level from Table 4. Give more explaination in discussion. What is the possible message that this trend can provide for the lake environmental management?

(4) It would be very interesting to compare DYRESM model with catchment model (e.g. Xinanjiang Model, SWAT) in the accuracy of inflow estimation if possible in this paper. 

Reviewer 2 Report

The article solves the problem of clarifying the components of the lake's water balance on the basis of observational data on hydrology and water quality. There is no scientific setting. Hydrological parameters are corrected for good agreement between natural and calculated values. Water quality data were used for the adjustment, which is highly questionable. There are no estimates of the accuracy of calculations for water quality parameters. There is no lake water quality model. Without a model, such calculations are not interesting. There are no new scientific results
